# Genome-Wide Identification of B-Box Gene Family and Candidate Light-Related Member Analysis of Tung Tree (*Vernicia fordii*)

**DOI:** 10.3390/ijms25041977

**Published:** 2024-02-06

**Authors:** Kai Shi, Guang Zhao, Ze Li, Junqin Zhou, Lingli Wu, Xiaofeng Tan, Jun Yuan

**Affiliations:** 1Key Laboratory of Cultivation and Protection for Non-Wood Forest Trees, Ministry of Education, Central South University of Forestry and Technology, Changsha 410004, China; shikai_csuft@163.com (K.S.); dickde@126.com (G.Z.); lize1853@163.com (Z.L.); zhoujunqin1234@126.com (J.Z.); wulingli0307@163.com (L.W.); 2Hunan Forestry Seedling Breeding Demonstration Center, The Forestry Department of Hunan Province, Changsha 410329, China

**Keywords:** tung tree, *VfBBX* gene family, light-shading treatment, gene expression, *VfBBX9*

## Abstract

Light is one of the most important environmental factors for plant growth. In the production process of tung oil tree cultivation, due to the inappropriate growth of shading conditions, the lower branches are often dry and dead, which seriously affects the yield of tung oil trees. However, little is known about the key factors of light-induced tree photomorphogenesis. In this study, a total of 22 *VfBBX* family members were identified to provide a reference for candidate genes in tung tree seedlings. All members of the *VfBBX* family have different numbers of highly conserved B-box domains or CCT domains. Phylogenetic evolution clustered the *VfBBX* genes into four categories, and the highest density of members was on chromosome 6. Interspecific collinearity analysis suggested that there were six pairs of duplicate genes in *VfBBX* members, but the expression levels of all family members in different growth and development stages of the tung tree were significantly divergent. After different degrees of shading treatment and physiological data determination of tung tree seedlings, the differential expression level and chlorophyll synthesis genes correlation analysis revealed that *VfBBX9* was a typical candidate nuclear localization transcription factor that was significantly differentially expressed in light response. This study systematically identified the *VfBBX* gene family and provided a reference for studying its molecular function, enhanced the theoretical basis for tung tree breeding, and identified excellent varieties.

## 1. Introduction

Light is an indispensable environmental factor for plants, playing a pivotal role in modulating their growth and developmental processes by regulating shade avoidance, directional growth, and photoperiodic flowering. In the widely utilized cultivation of plant seedlings in facilities, the facility skeleton and greenhouse film may exert a certain shielding effect on light, or changes in climate and weather conditions may induce shade, thereby influencing plant growth [1]. Research has demonstrated that plants exposed to partial shade or darkness typically exhibit a symptomatic shade-avoidance syndrome (SAS), characterized by weakened seed germination, elongated hypocotyls and petioles, reduced leaf surface area, etiolated leaves, accelerated flowering, and decreased branching [2]. These mechanisms reciprocally regulate various photomorphogenesis programs, impacting plant growth, fruit formation, and the production of secondary metabolites [3]. In the photomorphogenesis process, active photoreceptors negatively control the photomorphogenesis inhibitors, releasing downstream light-promoting transcription factors by absorbing visible light [4]. Shade-avoidance responses are mediated by photoreceptors, with typical nuclear-localized phytochromes promoting or inhibiting the degradation of various transcription factors, such as PHYTOCHROME-INTERACTING FACTORs (PIFs) [5], LONG HYPOCOTYL 5 (HY5) [6], and B-BOX (BBX) proteins [7].

The B-box (BBX) family represents a typical class of transcription factors involved in 26S proteasome-mediated degradation by directly activating E3 ubiquitin ligases, such as COP1 [7]. BBX family proteins typically contain one or two conserved BBX domains at the N-terminal region, and some members also possess a CCT (Constans, CO-like, and TOC1) structure at the C-terminal region [8]. In *Arabidopsis thaliana*, BBX proteins have been reported to exhibit diverse functions, including negative regulation of flowering time [9], involvement in brassinosteroid signaling and inhibition of seedling development [4], promotion of hypocotyl growth, primary root elongation, and UV-B tolerance [10]. Moreover, most BBX genes act as regulators of photomorphogenesis, influencing photomorphogenesis in *Arabidopsis* seedlings through interactions with COP1 or HY5. For instance, BBX32 plays a crucial role in the control of acclimation to high light in mature *Arabidopsis* leaves, and plants overexpressing BBX32 are strongly impaired in acclimation to intermittent episodes of high light [11]. BBX11 serves as a positive regulator of red light signaling, and the BBX11-BBX21-HY5 positive feedback loop promotes *Arabidopsis* photomorphogenesis [12]. BBX21, another positive regulator of light signaling, interacts with COP1 in vivo and depends on COP1 for 26S proteasome-mediated degradation in dark-grown *Arabidopsis* seedlings. Additionally, BBX20, a protein that positively regulates light signaling, interacts with COP1 and undergoes COP1-mediated degradation [13]. BBX21 binds to the T/G-box in the HY5 promoter, underscoring its role as a key component in the COP1-HY5 regulatory network [14]. Despite the well-established roles of BBX proteins in regulating the growth and development of Arabidopsis, their functions in forest tree model systems remain to be explored.

The Tung tree (*Vernicia fordii*) is a plant of the genus *Vernicia* in the family Euphorbiaceae. It is native to China and is an important industrial woody oil species known for its widespread distribution, ease of cultivation, and high oil production, making it one of the most extensively cultivated commercial tree species. As one of the most promising biomass energy tree species, tung oil tree seeds can be used as an environmentally friendly coating to meet the needs of the modern consumer market [15]. The Tung tree is a deciduous tree with a strong ability to adapt to its surroundings. It is light-loving, resistant to high temperatures and drought, and rapid growth and early fruiting are intricately linked to the process of photosynthesis. However, the rapid growth and expansive crown of the tung tree pose challenges, as the shade cast by the upper branches can hinder the development of lower branches, leading to a significant number of dead branches throughout the cultivation process. This situation directly impacts the light environment experienced by tung tree leaves, contributing to the occurrence of dead branches and adversely affecting the normal growth and yield of the tung tree [16]. However, existing research on tung tree photosynthesis primarily revolves around physiological comparative analyses of photosynthetic features in different varieties, with limited exploration into gene expression responses to the tung tree’s light environment.

To study the role of the BBX family in the process of light-induced photomorphogenesis during the growth and development of tung trees. In this study, *VfBBX* family members were identified from the genome sequence of the tung tree, which found that *VfBBX* transcription factors had a wide range of biological functions in the process of photomorphogenesis of tung tree seedlings. These findings can be used to provide fundamental guidance for the study of *VfBBX* gene functions, as well as offer a theoretical foundation for initiatives related to breeding, identification of superior varieties, and the development of high-quality cultivation practices for the tung tree.

## 2. Results

### 2.1. Genome-Wide Identification B-Box (BBX) Family Members in Tung Tree

Utilizing a hidden Markov model (HMM)-based profile associated with the B-box-type zinc-finger domain (PF00643), and putative BBX candidate genes were acquired through an HMMER3 search conducted on the tung tree protein database. A total of 22 *VfBBX* family members, were designated as *VfBBX1*–*VfBBX22* based on their positions on chromosomal scaffolds. The significant variation in molecular weights (Mw) spanned from 13,502.70 Da (VfBBX5) to 57,450.27 Da (VfBBX19). Isoelectric points (PI) of the VfBBX proteins demonstrated diversity, fluctuating between 4.40 (VfBBX21) and 7.52 (VfBBX5), signifying predominantly weakly acidic proteins. Subcellular localization prediction software indicated that these VfBBX proteins predominantly reside in the nucleus. Notably, only three members (VfBBX2, VfBBX19, and VfBBX21) exhibited extracellular localization, specifically in structures such as chloroplasts and mitochondria (Table 1).

### 2.2. Conserved Domains and Gene Structure Analysis of VfBBX Genes in Tung Tree

Conserved structural domains revealed that two distinctive forms of the conserved zinc finger domain of B-box, namely B-box1 (C-X2-C-X7–8-C-X2-D-X-A-X-L-C-X2-C-D-X3-H-X2 -N-X4-H, indicated in orange) and B-box2 (C-X2-C-X8-C-X7-C-X2- C-X4-H(N)-X6–8-H, indicated in green), were identified. Protein sequence alignment highlighted the high similarity in the conserved sequences of these two B-BOX domains. Additionally, the BBX family also exhibited a highly conserved CCT domain (R-X5-R-Y-X2-K-X3-R-X3-K-X2-R-Y-X2-R-K-X2-A-X2-R-X-R-X2-G-R-F-X-K), represented by a blue bar in Figure 1A. Notably, the cysteine (C) and aspartic acid (D) residues of the zinc finger exhibited substantial conservation within the B-BOX domain (Figure 1A). The conservative motif symbol was reversed due to the conserved amino acid residues (Asn, Leu, His, and Arg) in the B-box1 domain, making the B-box1 domain motifs more conserved than those of the B-box2 domain (Appendix A). Among the 22 VfBBX proteins, only three members (VfBBX2 and CaBBX21) contained a single B-box1 domain, while three proteins (VfBBX9, VfBBX10 and CaBBX18) possessed B-box1 and CCT domain. The remaining 17 genes featured both B-box1 and B-box2 domains, with 10 members including the conserved CCT domain. Gene structure analysis revealed that the number of exons in the *VfBBX* gene family ranged from one to five. Among them, 8 *VfBBX* genes (36.36%) had two exons, 3 members exhibited three exons, 7 *VfBBX* genes (31.81%) had four exons, and 3 members had five exons. Notably, *VfBBX5* was unique with only one exon (Figure 1B).

### 2.3. Chromosomal Distribution and Interspecies Synteny Analyses for VfBBXs

The chromosomal distribution analysis revealed that the 21 *VfBBX* genes were broadly distributed across 8 chromosomes, and *VfBBX1* was situated on chromosome ‘00’. Chromosome 7 exhibited the highest abundance of *VfBBX* genes, hosting 5 *VfBBX* members. Chromosomes 1 accommodated four genes, chromosomes 2 and 11 existed three members each, chromosomes 8 and 10 existed two *VfBBX* genes each, chromosomes 7 and 9 only accommodated one *VfBBX* each, while the remaining chromosomes contained no *VfBBX* member. A total of four *VfBBX* genes exist duplication genes phenomenon in the tung tree genome (Figure 2). It is noteworthy that *VfBBX2* and *VfBBX21* duplicate with *Vf07G0713*, yet *Vf07G0713* is not considered to be a member of the BBX family, indicating that there is still a loss of conserved regions during duplication. The evaluation of selection pressure in duplicated genes involved assessing non-synonymous mutations (Ka), synonymous mutations (Ks), and their ratio (Ka/Ks). The Ks values ranged between 1.86 and 1.94. For all duplicated gene pairs, the Ka/Ks values were less than 1.00, falling within the range of 0.16 to 0.28, suggesting that the sets of *VfBBX* duplicated genes underwent purifying selection throughout the evolutionary process. Among these, the duplicate gene pair *VfBBX21*-*Vf07G0713* exhibited the minimum Ks and the maximum Ka/Ks values, indicating a potentially stronger purifying selection acting on these two genes. However, due to the high sequence divergence value (pS > 0.75), the Ks value was not calculated for the *VfBBX7*-*VfBBX17 and VfBBX8*-*VfBBX17* duplicated gene pairs (Appendix A).

### 2.4. Cis-Elements Analysis in the Promoter of VfBBXs

Utilizing conserved protein sequences, the 22 VfBBX proteins were classified into four distinct phylogenetic groups. In the evolutionary classification, Group I included VfBBX4, VfBBX5, VfBBX3, VfBBX19, VfBBX14, VfBBX12, and VfBBX13 genes. Group II comprised VfBBX15, VfBBX16, VfBBX1, VfBBX10, and VfBBX11 members. Group III encompassed VfBBX2, VfBBX21, VfBBX7, VfBBX22, VfBBX6, VfBBX20, VfBBX8, and VfBBX17, while Group IV included VfBBX9 and VfBBX18 proteins (Figure 3A). Analysis of the promoter regions located 2000 base pairs (bp) upstream of the coding region was chosen to analyze cis-elements in the *VfBBX* genes using the PlantCARE website (Figure 4). Among these, light response regulation association cis-elements, such as Sp1, GATA-motif, GT1-motif, AE-box, G-box, GA-motif, and I-box. Moreover, some genes exhibited cis-elements associated with hormone-responsive elements such as ethylene response element (ERE), abscisic acid (ABA) responsive elements (ABRE, ABA responsiveness), and MeJA acid responsive elements (CGTCA-motif and TGACG-motif) were identified. Other stress-responsive elements included drought-related regulatory elements (MBS), low-temperature response elements (LTR, low-temperature responsiveness), anaerobic response regulatory elements (ARE, anaerobic responsiveness), defense and stress responsiveness (TC-rich motif), and wounding and pathogen responsiveness (W-box) (Figure 3B). Notably, each gene exhibited a unique composition of cis-elements (Appendix A), with *VfBBX1*, *VfBBX2*, *VfBBX9*, and *VfBBX21* specially containing a higher abundance of light-responsive cis-elements, while *VfBBX1* and *VfBBX2* displayed a greater prevalence of hormone-responsive elements.

### 2.5. Expressions Analyzed of VfBBXs in Divergence Stages by RNA-seq Data

The RNA-seq data analysis of *VfBBX* gene family expression across various developmental stages and organs in tung trees provided valuable insights into their potential functions (Figure 4A). There are some members specifically highly expressed in some tissues, for example, *VfBBX20* and *VfBBX3* were highly expressed in the 10 BF of male flower, indicating that these two genes may be significantly involved in the early stage of male flower formation. Some family members such as *VfBBX12*, *VfBBX4*, *VfBBX5*, *VfBBX14*, and *VfBBX19* may be involved in the whole process of endosperm formation, while *VfBBX10*, *VfBBX17*, *VfBBX20*, *VfBBX7*, and *VfBBX21* gene was specifically highly expressed in the roots and stems of tung tree. Particularly, *VfBBX9* was significantly highly expressed in the leaves of the tung tree. However, *VfBBX12*, *VfBBX4*, *VfBBX5*, *VfBBX10*, and *VfBBX17* displayed consistently low expression levels across various both male flowers and female flowers development stages, while *VfBBX20*, *VfBBX7*, *VfBBX9*, *VfBBX22*, *VfBBX15*, *VfBBX1*, *VfBBX16*, *VfBBX2*, and *VfBBX18* exhibited low expression levels throughout seed development stages. These findings underscore the diverse functions of *VfBBX* genes during the growth and development of tung trees across different tissues and developmental stages.

The assigned Gene Ontology (GO) terms of all 23 *VfBBX*s covered three main categories: cellular component, molecular function, and biological process based on a corrected *p*-value < 0.05. In the cellular component category, the highly enriched terms for all the *VfBBX* genes were the cell part and cell. For molecular function, *VfBBX*s were predominantly associated with the binding and the organic cyclic compound binding process. The biological process category showcased a wide range of assignments for *VfBBX* genes, including metabolic process, regulation of the biological process, and response to stress. Kyoto Encyclopedia of Genes and Genomes (KEGG) analysis showed all *VfBBX* gene family members significantly enriched in metabolic pathway, biosynthesis of secondary metabolism, fatty acid biosynthesis, plant circadian rhythm, ribosome, photosynthesis, and flavonoid biosynthesis. GO and KEGG terms provide valuable insights into the diverse functional roles played by the *VfBBX* gene family across various processes.

### 2.6. Different Light-Shading-Treatment of Tung Tree Seedlings

To study the role of the *VfBBX* family and their involvement in the photomorphogenesis of the tung tree, the seedlings were treated with normal light (L1), 75% (L2), 50% (L3), and 25% (L4) light intensity levels, respectively. As shown in Figure 5A and Appendix A, the average plant height of L3 treatment was significantly higher than that of seedlings under L1, which was significantly increased by 35.66%, 56.23%, and 45.05%, respectively, compared with L1, L2, and L4 (*p* < 0.05), indicating that 50% light led to overgrowth of seedlings. L4 treatment directly affected seedling dwarfing and leaf thinning. Further, tung tree seedlings exhibit an increasing trend in chlorophyll a, chlorophyll b, and total chlorophyll content, while the chlorophyll a/b ratio gradually decreases with decreasing light intensity. Chlorophyll a, chlorophyll b, and total chlorophyll reached their maximum in the L4 treatment, which significantly increased compared to L1 (Figure 2B). The peroxidase (POD) activity increased initially and then decreased with the increasing shading intensity. The L3 treatment showed a significant elevation in POD activity compared to other treatments, with increases of 83.61%, 52.44%, and 41.90% relative to the other L1, L2, and L4 treatments, respectively (Figure 5C). However, the MDA content, superoxide dismutase (SOD) activity and catalase (CAT) activity in tung tree seedlings significantly increased with the increase in shading intensity. The MDA content in L4 increased significantly by 87.66%, 82.50%, and 45.49% compared to L1, L2, and L3, respectively. The CAT activity in the L4 treatment exhibited a significant increase of 56.18%, 26.87%, and 7.02% compared to the L1, L2, and L3 treatments, respectively (Figure 5D,E). In addition, the SOD activity of the L4 treatment increased with significant differences by 49.33%, 40.56%, and 7.73% compared to the L1, L2, and L3 treatments, respectively (Figure 5F).

Moreover, qRT-PCR analysis to validate the expression patterns of *VfBBX*s in the different light-shading treatments. According to the expression level, six genes including *VfBBX4*, *VfBBX5*, *VfBBX9*, *VfBBX12*, *VfBBX13,* and *VfBBX22* were found to be specifically highly expressed in leaves as candidate shading-responsive genes. Results suggested that the expression level of *VfBBX5*, *VfBBX12,* and *VfBBX13* were significantly decreased only under L2 treatment, but no significant changes under both L3 and L4 treatment. The expression level of *VfBBX4* was significantly high under the L3 treatment, and *VfBBX22* has a high expression level under the L4 treatment. In particular, the expression level of *VfBBX9* was significantly high under both L2 and L4 treatment (Figure 5G).

### 2.7. A Candidate VfBBX9 May Be Involved in Photomorphogenesis Process

Due to the effect of light on chlorophyll content, Pearson correlation analysis was performed between chlorophyll biosynthesis genes and *VfBBX* family members. *VfBBX9* and *VfBBX22* were strongly positively linked with most chlorophyll biosynthesis genes, and *VfBBX9* had the highest expression in leaves according to gene expression. As a result, we believe *VfBBX9* is an important candidate gene controlling chlorophyll biosynthesis and photomorphogenesis (Figure 6A and Appendix A). The transcriptional activity of *VfBBX9* was studied in a yeast system to better understand its function. The Y2H Gold yeast strain expressing BD- *VfBBX9* and the activation domain (AD) thrived on SD-Trp/-Ade/-His media similarly to the positive control, indicating that *VfBBX9* possesses robust transcriptional activation activity (Figure 6B). Subcellular localization was accomplished by a method using *35S::*VfBBX9-eGFP transiently transformed tobacco leaves. Green eGFP fluorescence was observed throughout the cytoplasm and nucleus in *35S::*eGFP transiently transformed control tobacco cells by confocal laser microscopy. In contrast, *35S::*VfBBX9-eGFP only 1detected green fluorescence in the nucleus. Therefore, *VfBBX9* is considered a typical nuclear protein (Figure 6C).

## 3. Discussion

Light plays a pivotal role in regulating diverse photomorphogenic processes, encompassing plant shade avoidance, directional growth, and photoperiodic flowering. These light-induced processes have significant implications for plant growth, fruit development, and the synthesis of secondary metabolites. The B-box (BBX) transcription factors, constituting a subset of the zinc finger protein family, have garnered considerable research interest due to their diverse biological functions, especially in the light-photomorphogenesis process [17]. This study identified a total of 22 *VfBBX* family members in the tung tree, and the distribution of *VfBBX*s was uneven across the 8 chromosomes of the tung tree genome. The members of *BBX* were comparatively lower than that of *Arabidopsis* (32 *ATBBX*s) [18], and 29 tomato *SlBBX*s [19] (Appendix A). Differences in the numbers of BBX family transcription factors among species may arise from species-specific duplications or deletions during the course of evolution. Furthermore, the subcellular location of *VfBBX* family genes implies that the majority of them are localized in the nucleus and are highly likely to be transcription factors, which is consistent with the findings in other species [20]. The B-box conserved motifs at the C-terminal were crucial for protein interaction and transcriptional regulation, and the N-terminal was complemented by the CCT domain, which is primarily involved in nuclear transport and transcriptional regulation [8]. Furthermore, the exon-intron structures contributed to gene functions, with the *VfBBX* family displaying variations in exon numbers ranging from two to five. Functional diversity also arose from distinct cis-regulatory elements in the promoter regions [21]. Numerous cis-elements in the promoter region were associated with light responsiveness, such as Box 4, G-Box, TCT-motif, and GT1-motif. Several BBX proteins, like AtBBX20, are known to play pivotal roles in light signaling pathways [22]. Additionally, cis-elements like ABRE, TC-rich repeats, TCA-element, and ARE were implicated in abscisic acid responsiveness or anaerobic induction. Hormone-related cis-elements, including ERE, CGTCA-motif, TGACG-motif, and GARE-motif, were also present in the *VfBBX* promoter region. The diverse domains, structures, and cis-elements within the promoter region collectively contribute to the varied functions exhibited by the *VfBBX* family of transcription factors.

Gene duplication stands out as a pivotal mechanism driving the expansion and functional diversification of gene families [23]. Within the tung tree, six duplicated pairs were identified, contributing significantly to the enlargement of the *VfBBX* family. This pattern aligns with findings in other plant species such as the pepper *CaBBX* family [8], and tomato *SlBBX* family [19]. Intriguingly, the expression levels of the members of the *VfBBX* family members vary significantly despite the fact that many of their gene pairs are duplicated. It is reasonable to speculate that functional transcription and translation areas have developed distinct functions, while duplicated genes typically repeat non-specific conserved non-functional sequences (Figure 4). Furthermore, the *VfBBX* family genes show a wide range of differential expression in each growth and development stage of the tung tree, indicating that it may be a crucial transcription factor family engaged in the entire life cycle of the tung tree. Studies have shown that *BBX* genes have divergence functions in other species, such as *SlBBX20* was found to be a positive regulator of fruit carotenoid accumulation, which interacted with SlDET1 leading to the ubiquitination and 26S proteasome-mediated degradation of SlBBX20 in tomato [24]. Furthermore, overexpressed *SlBBX17* has affected plant growth and enhanced heat tolerance in tomatoes [25]. However, the functions of BBX proteins have rarely been found in other fruit crops except the fleshy fruits tomato model system.

More importantly, a number of studies have revealed that BBX proteins are involved in light signaling and either promote or inhibit photomorphogenic development. In *Arabidopsis*, BBX21, a positive regulator of light signaling, interacted with COP1 in vivo and relies on COP1 targeting 26S proteasome-mediated degradation in dark-grown *Arabidopsis* seedlings [14]. HY5 contributes to the degradation of BBX22 under light, and the expression of BBX22 will accumulate briefly during photomorphogenesis, indicating that BBX22 is also a positive regulator [26]. BBX20, a protein that positively regulates light signaling, interacts with COP1 and undergoes COP1-mediated degradation [13]. On the contrary, BBX28 can interfere with the binding of transcription factor HY5 to its target gene promoter and negatively regulate photomorphogenesis [27]. Similarly, *BBX25* and *BBX24*, as transcriptional co-repressors, may regulate light-mediated seedling development by forming inactive heterodimers with HY5 and down-regulating the expression of *BBX22* [28]. These studies have also extensively revealed that BBX proteins can be degraded by ubiquitin-mediated protein degradation in the light signal transduction pathway, thereby affecting their regulatory activity. We also determined the expression levels of BBX family members following various shading treatments on tung tree seedlings. Simultaneously, based on the expression level and link between chlorophyll biosynthesis genes and the *VfBBX* family, we hypothesized that *VfBBX9* could be a possible component implicated in the high response of tung tree seedlings to light. We also demonstrated that *VfBBX9* is a nuclear localization transcription factor with high transcriptional activation activity and that it shares the most homolog with BBX14 of *Arabidopsis*. BBX14 protein is considered a negative regulator that inhibits nitrogen deprivation and dark-induced leaf senescence [29]. This study provides a candidate target for the subsequent study into the function and light response of the tung tree *VfBBX* family.

In conclusion, in this study, we included the fundamental data of the initial members of the tung tree *VfBBX* family genes. The expression of *VfBBX* genes in tung tree seedlings under various light-shading treatments and correlation analysis of chlorophyll biosynthesis genes suggests that *VfBBX9*, as a nuclear localization gene, maybe a candidate critical transcription factor involved in the photomorphogenesis of tung tree. The next study will look into the biological function and regulatory network of *VfBBX9* as a potential role in the light response of tung trees. This study not only provides comprehensive and accurate identification information for the *BBX* gene family of tung trees but also provides a theoretical foundation for the high-yield and high-quality cultivation of tung trees.

## 4. Materials and Methods

### 4.1. Identification of VfBBX Transcription Factors in Tung Tree

To discern the BBX family of transcription factors, the hidden Markov model (HMM) profile of the B-box-type zinc finger domain (PF00643) was used and sourced from the Pfam database [30]. The reference genomes of the tung tree (*Vernicia fordii*) constituted the basis of this study [15]. Employing the HMMER SEARCH3.0 with a cutoff E value ≤ 0.01 and SMART database, the presence of conserved domains is validated within the BBX family members [31]. Theoretical isoelectric point (pI) and molecular weight (Mw) values were calculated using the ExPaSy online tool (https://web.expasy.org/compute_pi/, accessed on 10 October 2023) [32]. For protein modeling, the SWISS-MODEL online tool (https://swissmodel.expasy.org/, accessed on 10 October 2023) was employed [33], while the putative subcellular localization of genes was predicted using the Softberry online software (http://www.softberry.com, accessed on 11 October 2023).

### 4.2. Conserved Motifs, Gene Structures, and Phylogenetic Analysis of VfBBX Members

The identification of conserved motifs was carried out using the MEME online tool (Version 5.1.0, National Institutes of Health, Bethesda, MD, USA), and the aligned domains were visualized using DNAMAN (Version 8.0.8, Lynnon Biosoft, San Ramon, CA, USA). Predictions regarding the gene structures of *VfBBX*s were made using the Gene Structure Display Server [34]. The construction of an unrooted neighbor-joining phylogenetic tree for VfBBXs, SlBBXs, and AtBBXs proteins, along with bootstrap testing replicated 1000 times, was performed using MEGA11 (https://www.megasoftware.net/home) [35].

### 4.3. Synteny Analysis and Cis-Elements Analysis of VfBBXs

Intra-species collinearity analysis and synonymous (Ks) and nonsynonymous (Ka) substitution rates are used by Tbtools-II [36]. The Circos diagram illustrates the chromosomal location, collinearity, and Ka/Ks values for the genes. The promoter region, selected as 2000 bp upstream sequences of coding DNA sequences (CDS), was analyzed for cis-elements using the PlantCARE web tool (http://bioinformatics.psb.ugent.be/webtools/plantcare/html/, accessed on 20 October 2023) [37].

### 4.4. Materials and RNA-seq Analysis

The experiment was carried out at Central South University of Forestry and Technology (28.10° N, 113.23° E), which belongs to a subtropical monsoon climate, and the frost-free period of the whole year is about 275 days. The mean annual temperature ranges from 28 to 37.5 °C, and the 88 mm average monthly precipitation. The material was collected from the National Germplasm Resource Storage of *Vernicia fordii* in Qingping Town, Xiangxi, Hunan, which is located at 110°29′ E, 28°32′ N, with an altitude of about 530–600 m. The indoor shade was dried for three months, and the tung tree seedlings were used after 1 month of seed germination. The raw RNA-seq data utilized in the analysis were obtained from the Tung Tree eFP Browser [15]. GO enrichment analyses were carried out using the Plant Transcriptional Regulatory Map [38], with a significance threshold set at a corrected *p*-value < 0.05. KEGG analyses use KOBAS-i:intelligence online tools (http://bioinfo.org/kobaskobas3/?t=1, accessed on 25 October 2023) and plot by R [39].

### 4.5. Different Light-Shading Treatment Analysis of Tung Tree Seedlings

A total of four light treatments were set up in this study, and the light transmittance was measured by LI-6400 portable photosynthesis meter. The optical density was controlled by different layers of black shading net. The relative light intensities of the four treatments were 100% (L1), 75% (L2), 50% (L3), and 20% (L4), respectively. The seedlings of tung tree with basically the same size and growth were transplanted into a plastic basin with an inner diameter of 24 cm and a depth of 23 cm (1/2 of nutrient soil and quartz sand), 18 pots for each treatment. A total of 72 pots were treated with four treatments, and a shading test was carried out after 40 days of normal water management. The seedling height, leaf area, and ground diameter of tung tree seedlings of four treatments were measured by vernier caliper and tape ruler. Nine plants were selected from each treatment and the mean ± SDs value was taken. The chlorophyll content, Total-Superoxide Dismutase (T-SOD), Catalase (CAT), Peroxidase (POD), and oxidative products including malondialdehyde (MDA) were quantified of tung tree seedlings by the plant assay kits (Nanjing Jiancheng Biotechnology Co., Ltd., Nanjing, China). Two grouping-variable bar graphs and a one-way analysis of variance (ANOVA) were employed. Data are presented as means ± standard deviation (SD; *n* = 9).

### 4.6. Quantitative Real-Time PCR (qRT-PCR) Analysis

Total RNA extraction was performed using the TransZol kit (TransGen Biotech, Inc., Beijing, China), and cDNA synthesis was carried out using the HiScript^®^ II QRT SuperMix for qPCR (+gDNA wiper) kit from Vazyme (Piscataway, NJ, USA). The LightCycle * 96 Real-Time PCR System from Roche (Basel, Switzerland) was employed for qRT-PCR experiments. A 50 μL reaction system was configured, and each sample underwent three biological repeats with three technical repeats. The HiScript^®^ II Q RT SuperMix for qPCR (+gDNA wiper) kit from Vazyme (Piscataway, NJ, USA) was utilized for reverse transcription. Relative expressions were calculated using the 2^−ΔΔCt^ formula. Primers for *VfBBX*s and the actin *EF1α* gene are detailed in Appendix A. All data are expressed as mean ± SDs (Student’s *t*-test; * *p* < 0.05, ** *p* < 0.01, and *** *p* < 0.001 considered statistically significant).

### 4.7. Pearson Correlation Coefficient and Transcriptional Activity Analysis

The Pearson correlation coefficient (r) method was used to calculate the correlation between chlorophyll biosynthesis genes and *VfBBX* transcription factors. The larger the absolute value of r is, the stronger the correlation is, and the absolute value of r is greater than 0.5 showed in Figure 6A. The full-length CDS of *VfBBX9* was inserted into the pGBDKT7 vector. The pGBDKT7-VfBBX9, pGBDKT7-53 vector (positive control), and pGBDKT7-lam vector (negative control) were transformed into the Y2HGold yeast strain. Three SD/-Trp and SD/-Trp/-His/-Ade plates were used to culture the altered yeast cells. Primer information is provided in Appendix A.

### 4.8. Subcellular Localization Assay of VfBBX9

The complete coding sequence (CDS) of *VfBBX9*, excluding the stop codon, was cloned into the pSUPER1300 vector and fused with the green fluorescent protein (GFP) to generate a *VfBBX9*–GFP fusion protein. The empty pSUPER1300 vector served as the control. These constructs were then introduced into the *Agrobacterium tumefaciens* strain GV3101. Co-transformation of the GFP-fusion constructs and the nuclear location marker (NF-YA4-mCherry) was performed in *Nicotiana benthamiana* leaves. Following 48 h of infiltration, fluorescence signals were observed at excitation/emission wavelengths of 488/510 nm (GFP) and 552/610 nm (mCherry) using a confocal laser scanning microscope (Leica SP8, Heidelberg, Germany). Primer information is provided in Appendix A.

## Figures and Tables

**Figure 1 ijms-25-01977-f001:**
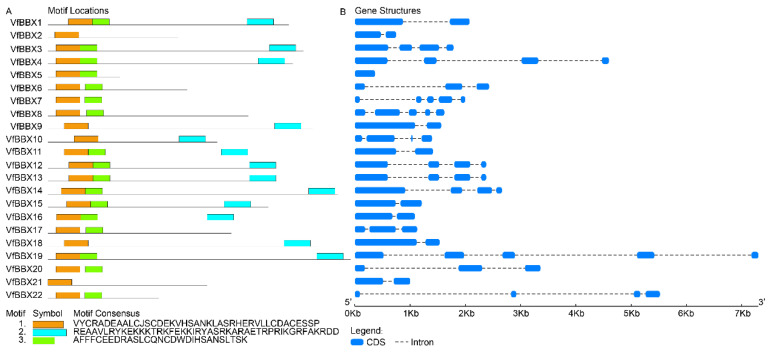
Conserved domains and gene structure of the *VfBBX* gene family. (**A**) The distributions of conserved motifs in *VfBBX*s, orange bars represent as B-box1 motif, blue bars represent as B-box2 motif, and green bars represent as CCT motif; (**B**) Gene structure of *VfBBX* genes, the blue squares represent the exon, and the black dotted lines represent intron.

**Figure 2 ijms-25-01977-f002:**
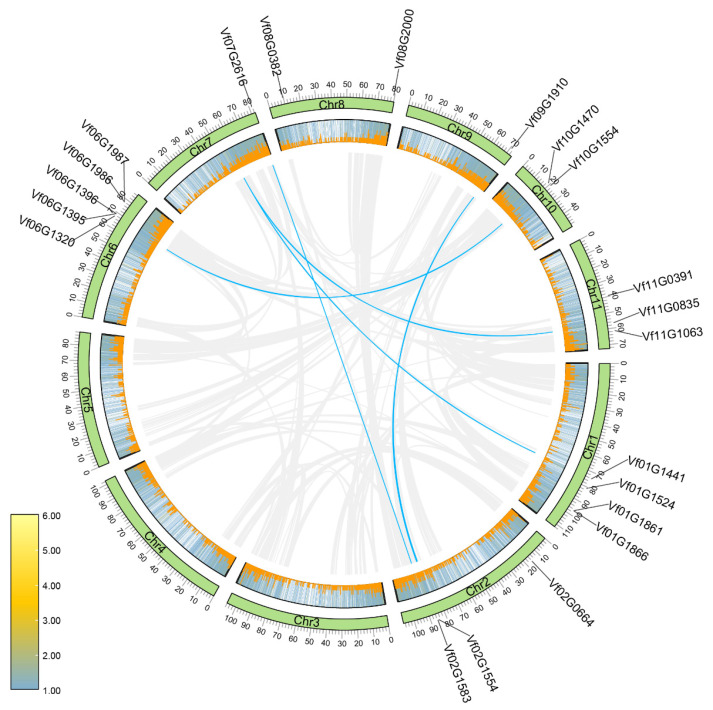
Chromosomal location and interspecies gene synteny in *VfBBX* genes. Blue lines in the middle indicate duplicated gene pairs of *VfBBX*s. The positions of *VfBBX* genes in the tung tree genome are marked on chromosomes; Chr refers to chromosomes.

**Figure 3 ijms-25-01977-f003:**
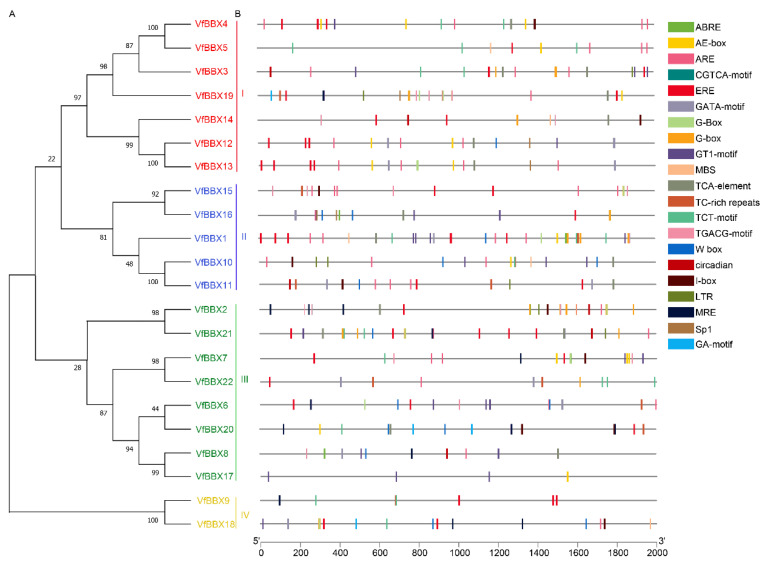
Phylogenetic relationships and predicted cis-elements analysis of *VfBBX* genes. (**A**) Phylogenetic tree for the 22 *VfBBX*s. Red genes were classified in Group I, blue members belonged to Group II, green genes to Group III, and yellow members to Group IV. (**B**) Promoter sequences (−2000 bp) of VfBBXs were analyzed using PlantCARE. Different shapes and colors represent different elements. Annotated cis-elements are listed in Appendix A.

**Figure 4 ijms-25-01977-f004:**
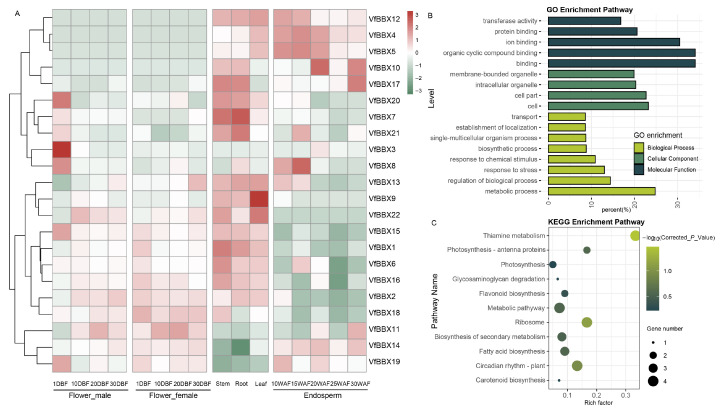
Expression patterns, GO, and KEGG enrichment of *VfBBX* family members. (**A**) Expression profiles based on RNA-seq fragments per kilobase million (FPKM) data. WAF: weeks after flowering; DBF: days before flowering; (**B**) GO enrichment pathway analysis of *VfBBX* genes. Different degrees of the green bar represented cellular components, molecular function, and biological process, respectively; (**C**) KEGG enrichment analysis of *VfBBX* family members.

**Figure 5 ijms-25-01977-f005:**
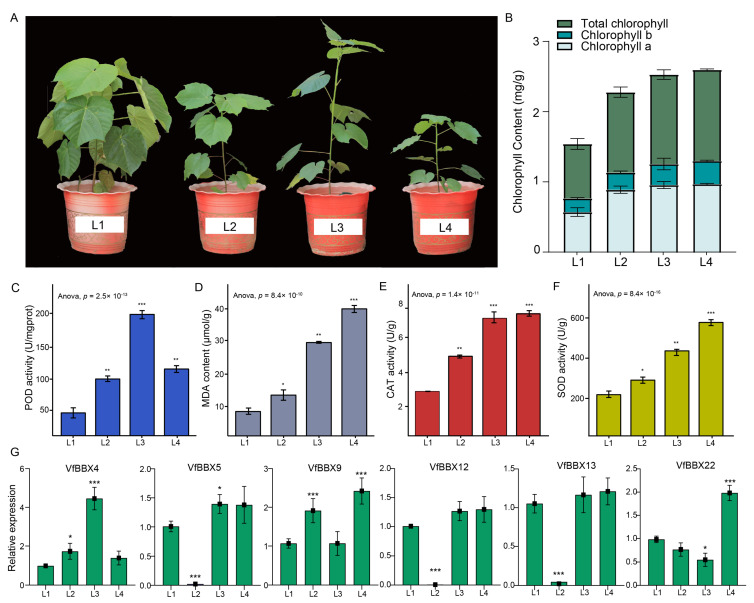
Phenotypes of different light-shading treatments of tung tree seedlings. (**A**) Tung tree seedlings in different light-shading treatments, normal light (L1), 75% (L2), 50% (L3), and 25% (L4) light intensity levels; (**B**) Chlorophyll content of tung tree seedlings in different light-shading treatments; (**C**) POD activity of tung tree seedlings in different light-shading treatments; (**D**) MDA content of tung tree seedlings in different light-shading treatments; (**E**) CAT activity of tung tree seedlings in different light-shading treatments; (**F**) SOD activity of tung tree seedlings in different light-shading treatments, * *p* < 0.05, ** *p* < 0.01, and *** *p* < 0.001 considered statistically significant; (**G**) Relative expression analysis of candidate *VfBBX* genes in different light-shading treatments, * *p* < 0.05, and *** *p* < 0.001 considered statistically significant.

**Figure 6 ijms-25-01977-f006:**
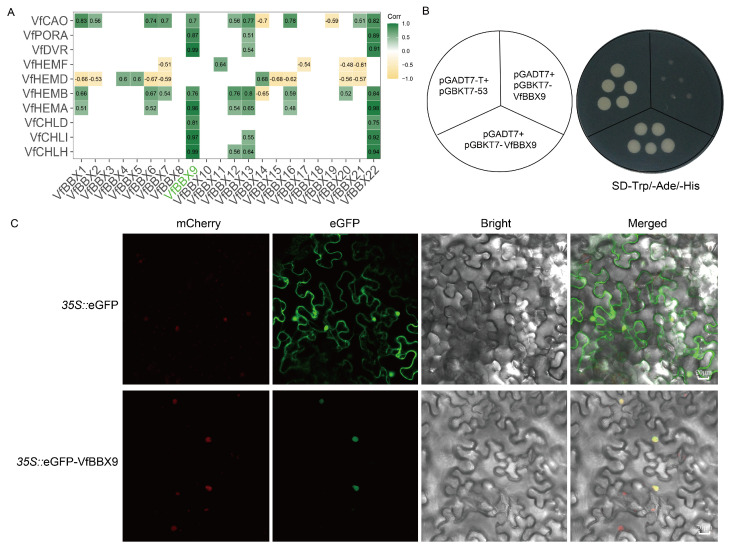
Candidate gene analysis, transcriptional activity, and subcellular localization of *VfBBX9*. (**A**) Pearson correlation analysis between chlorophyll biosynthesis genes and *VfBBX* genes, the larger the |r| value, the higher the correlation, and the correlation of |r| > 0.45 is shown in the figure. The positive correlation is represented by green, and the negative correlation is represented by yellow; (**B**) Transcriptional activation activity of *VfBBX9* in yeast. Positive control, BD-53 + AD-T; negative control, pGBKT7-VfBBX9 + pGADT7; (**C**) The fusion protein *35S::*GFP-VfBBX9 and the control vector was transiently expressed in tobacco leaves and subsequently observed using fluorescence microscopy. The scale bar represents 20 μm.

**Table 1 ijms-25-01977-t001:** The basic information of *VfBBX* gene family in tung tree.

ID	GeneID	Chr	Start	End	Strain	PI	Mw (Da)	Subcellular Localization
*VfBBX1*	*Vf00G1104*	0	89,662	91,738	−	6.01	45,087.65	Nuclear
*VfBBX2*	*Vf01G1441*	1	74,174,775	74,175,523	−	4.82	24,310.77	Extracellular
*VfBBX3*	*Vf01G1524*	1	80,724,519	80,726,304	+	5.14	48,109.19	Nuclear
*VfBBX4*	*Vf01G1861*	1	96,727,792	96,732,389	−	5.26	45,193.41	Nuclear
*VfBBX5*	*Vf01G1866*	1	96,872,982	96,873,350	−	7.52	13,502.70	Nuclear
*VfBBX6*	*Vf02G0664*	2	16,076,935	16,079,364	+	4.91	26,074.67	Nuclear
*VfBBX7*	*Vf02G1554*	2	84,978,268	84,980,266	+	6.30	23,345.27	Nuclear
*VfBBX8*	*Vf02G1583*	2	85,667,653	85,669,272	−	5.12	38,733.10	Nuclear
*VfBBX9*	*Vf06G1320*	6	67,314,277	67,315,841	−	5.64	51,051.01	Nuclear
*VfBBX10*	*Vf06G1395*	6	69,031,670	69,033,068	+	4.77	32,524.05	Nuclear
*VfBBX11*	*Vf06G1396*	6	69,040,070	69,041,484	+	5.61	39,894.44	Nuclear
*VfBBX12*	*Vf06G1986*	6	77,933,550	77,935,926	−	5.80	43,554.53	Nuclear
*VfBBX13*	*Vf06G1987*	6	77,938,312	77,940,688	−	5.80	43,538.49	Nuclear
*VfBBX14*	*Vf07G2616*	7	81,738,810	81,741,471	−	6.52	54,454.33	Nuclear
*VfBBX15*	*Vf08G0382*	8	9,584,101	9,585,313	−	6.35	41,569.61	Nuclear
*VfBBX16*	*Vf08G2000*	8	79,397,606	79,398,694	−	5.94	37,033.50	Nuclear
*VfBBX17*	*Vf09G1910*	9	70,869,957	70,871,084	+	6.35	34,220.51	Nuclear
*VfBBX18*	*Vf10G1470*	10	18,835,584	18,837,119	+	5.10	51,746.70	Nuclear
*VfBBX19*	*Vf10G1554*	10	20,915,531	20,922,830	+	5.91	57,450.27	Extracellular
*VfBBX20*	*Vf11G0391*	11	39,393,706	39,397,065	+	5.79	31,878.61	Nuclear
*VfBBX21*	*Vf11G0835*	11	56,711,530	56,712,533	−	4.40	29,473.33	Extracellular
*VfBBX22*	*Vf11G1063*	11	61,707,357	61,713,797	−	5.31	20,704.38	Nuclear

## Data Availability

Data is contained within the article and Appendix A.

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
