# Peer review of "Genome-Wide Identification of B-Box Gene Family and Candidate Light-Related Member Analysis of Tung Tree (Vernicia fordii)"

_ijms, 2024, doi:10.3390/ijms25041977_

Round 1

Reviewer 1 Report

Comments and Suggestions for Authors

The article deals with the current and current issue of the influence of light on the physiological manifestation of an economically important plant species for oil production. In the introduction, the correct issue of crown development in tree-fruit species, important for research, is presented. This reasoning is that the development of tree species at later age stages will cause poorer light penetration and shading within and in the lower parts of the canopy. This ultimately leads to a reduction in the photosynthetic and economic production performance of these areas of the plant. The correctly constructed initial hypothesis is appropriately followed up in the chapter on materials and methods, where the correct testing methods and genome-wide identification of candidate genes are chosen. VfBBX family members were identified from the genome sequence of tung tree in the resulting interpretation. VfBBX transcription factors had a wide range of biological functions in the process of photomorphogenesis of tung tree seedlings. Pearson correlation analysis between chlorophyll biosynthesis genes and VfBBX genes was chosen appropriately for interpretation.

I suggest the following to the authors for improvement:

In the introductory part, more thoroughly describe the Tung Tree (Vernicia fordii) starting test material. Better describe the origin and initial genotypic characteristics of the population. This will allow the reader to have a better idea of the starting plant material and possibly better repeatability of the results.

Since this is basic research, I recommend adding to the discussion in one or two sentences a more concrete idea of how the findings of the evaluated work will be applied in subsequent applied research. So far, their expression is too general and not very specific.

In conclusion, I state that the article is suitable for publication after minor corrections. This is scientific quality basic research that can be used for further follow-up applications.

Reviewer 2 Report

Comments and Suggestions for Authors

Basically, the manuscript's topic corresponds to the journal's topic, and it is prepared properly.

Judging from the introduction, the manuscript is relevant for the field and original enough.

This study not only systematically identified the VfBBX gene family and provided a reference for studying its molecular function, but also provides a candidate transcription factor that may be involved in the photomorphogenesis of tung tree, and enhanced theoretical basis for tung tree breeding and identification of excellent varieties.

References are appropriate. In my opinion, the tables and figures are prepared properly.

However, there are a few notes:

1. In this manuscript, a total of 22 VfBBX family members were identified to provide a reference for candidate genes of tung tree seedlings. But a clearly formulated research objective should be provided at the end of the introduction.

2. Although conclusions are not mandatory, I suggest adding them at the end of the manuscript. Many results were analyzed, so the most important ones could be highlighted in the conclusions.

3. The methodology is described in detail. I don't have any major comments. Editorial observation: line 394 - it is written: “The average temperature was 28-37.5 °C…”. If you write “average”, it should be a single number. If you want to specify temperature ranges, you should not write “average”.

Reviewer 3 Report

Comments and Suggestions for Authors

The manuscript presents molecular studies leading to identification of the B-box (BBX) family of genes in tung tree (Vernicia fordi) genome, which may correspond to transcriptional factors implicated in plant morphogenesis. 

A large, original and valuable work has been done in the respect, but I do not feel competent to evaluate the correctness of the analysis. 

My critical reservations apply to the experimental design and presenting data which led the authors to the conclusion that members of the VFBBX gene family are involved in the molecular control of the plant response to light (precisely, to shade). 

This is a crucial thesis and thus it should be convincingly tested and proved.

The authors performed experiment with four light intensity levels. The results are documented via photos of selected plants, only. A reader can not image biological variability (via photos or presented as a figure for the seedling length, leaf number) to properly evaluate the physiological response.

Surprisingly, despite the imposed factor varied quantitatively we did not observe any trend in gene expression date, but rather random variations (often dramatic drop value at mild stress) along with the shade increase, that can not be obviously explained. This may indicate methodical flaws (e.g. sampling pattern). 

Moreover use of data with strong outliers (VFBBX5, VFBBX12, VFBBX13) for calculation of correlation coefficients is not recommended since results are almost always false. 

Further, the number of pairs (points) used to test a relationship is necessary as allows a reader to evaluate the power of statistical test, not only its statistical significance. 

Generally, statistics is a weak side of the manuscript and is not described sufficiently. Student's t-test (Fig. 5) is not appropriate for the problem.

I can not explain the paradox observed in the results, where the gene expression values of VfBBX9 and VfBBX22 provide very similar correlation coefficients with chlorophyll synthesis genes , while they dramatically vary in dependence on the shade level between each other (Fig. 6a). On the other hand, the expression of VfBBX5 and VfBBX13 take a similar pattern across the shade levels, while the correlation coefficients are completely different. In my opinion, these date do not allow to draw any reasonable conclusion related to the plant response to the shade.

It could be helpful for a reader to provide (as a supplement), figures presenting the levels of chlorophyll synthesis genes expression and the relationships between quantities used for the correlation analysis (as scatter plots) with precise methodical description.

Concussion presented in the abstract and discussion seems to be exaggerated taking into account of the evidence strength.

Additional notes:

line 28

Keywords: .... :expression".............  - gene expression

line 219 and elsewhere

"normal light (L1), 75 % shading (L2), 50 % shading (L3) and 25 % shading (L4)"......

sounds as if the 75% shading corresponded to a heavy shading while the 25% shading to mild shading. I seems to be confusing and is not consistent with line 408.

I would use 25%, 50% and 75% shade or 75%, 50%, 25% light intensity levels.

Round 2

Reviewer 3 Report

Comments and Suggestions for Authors

The manuscript has been improved by the authors in several points that render it acceptable for publication in the jurnal. Still english correction seems to be necessary.